# Pervasive structural heterogeneity rewires glioblastoma chromosomes to sustain patient-specific transcriptional programs

Ting Xie [1], Adi Danieli-Mackay[1], Mariachiara Buccarelli[2], Mariano Barbieri[1], Ioanna Papadionysiou[1], Q. Giorgio D'Alessandris[3,4], Claudia Robens [5], Nadine Übelmesser [1], Omkar Suhas Vinchure[6], Liverana Lauretti[3], Giorgio Fotia[7], Roland F. Schwarz [5,8], Xiaotao Wang [9,10], Lucia Ricci-Vitiani[2], Jay Gopalakrishnan[6,11], Roberto Pallini[3] ✉ & Argyris Papantonis [1] ✉

Glioblastoma multiforme (GBM) encompasses brain malignancies marked by phenotypic and transcriptional heterogeneity thought to render these tumors aggressive, resistant to therapy, and inevitably recurrent. However, little is known about how the spatial organization of GBM genomes underlies this heterogeneity and its effects. Here, we compile a cohort of 28 patient-derived glioblastoma stem cell-like lines (GSCs) known to reflect the properties of their tumor-of-origin; six of these were primary-relapse tumor pairs from the same patient. We generate and analyze 5 kbp-resolution chromosome conformation capture (Hi-C) data from all GSCs to systematically map thousands of standalone and complex structural variants (SVs) and the multitude of neoloops arising as a result. By combining Hi-C, histone modification, and gene expression data with chromatin folding simulations, we explain how the pervasive, uneven, and idiosyncratic occurrence of neoloops sustains tumor-specific transcriptional programs via the formation of new enhancer-promoter contacts. We also show how even moderately recurrent neoloops can relate to patient-specific vulnerabilities. Together, our data provide a resource for dissecting GBM biology and heterogeneity, as well as for informing therapeutic approaches.

Glioblastoma multiforme (GBM) tumors that are wild-type for the *IDH* gene constitute the most frequent brain malignancy in adults[1]. Despite surgical resection, GBMs resist chemotherapy, inevitably recur, and are highly invasive. Hence median patient survival is ~15 months from the time of diagnosis[2], and therapeutic options at recurrence remain scarce[3,4]. These are all attributed to the genomic[5–7], epigenomic[8–10], and transcriptional heterogeneity of GBMs[8,9].

In normal tissue, the three-dimensional (3D) organization of chromosomes coordinates gene activation and repression to give rise to homeostatic transcriptional programs[11–13]. This 3D organization is

[1]Institute of Pathology, University Medical Center Göttingen, Göttingen, Germany. [2]Department of Oncology and Molecular Medicine, Istituto Superiore di Sanità, Rome, Italy. [3]Department of Neuroscience, Catholic University School of Medicine, Rome, Italy. [4]Department of Neuroscience, Fondazione Policlinico Universitario A. Gemelli IRCCS, Roma, Italy. [5]Institute for Computational Cancer Biology (ICCB), Center for Integrated Oncology (CIO), Cancer Research Center Cologne Essen (CCCE), University of Cologne, Cologne, Germany. [6]Institute of Human Genetics, University Hospital and Heinrich-Heine-University Düsseldorf, Düsseldorf, Germany. [7]Centre for Advanced Studies, Research and Development in Sardinia (CRS4), Pula, Italy. [8]Berlin Institute for the Foundations of Learning and Data (BIFOLD), Berlin, Germany. [9]Institute of Reproduction and Development, Fudan University, Shanghai, China. [10]Research Units of Embryo Original Diseases, Chinese Academy of Medical Sciences, Shanghai, China. [11]Institute of Human Genetics, Jena University Hospital and Friedrich Schiller University of Jena, Jena, Germany. ✉e-mail: roberto.pallini@unicatt.it; argyris.papantonis@med.uni-goettingen.de

disrupted at multiple levels in the context of human disease, including cancer[14–16]. Structural (SVs) and copy number variants (CNVs) in tumor cells can rewire the 3D genome in ways that allow for the aberrant activation of oncogenes[17,18] or the repression of tumor suppressors[19]. For example, deletion of a boundary insulating two neighboring topologically-associating domains (TADs)[20] can lead to aberrant interactions between an oncogene in one TAD and active enhancers in the other, a phenomenon known as "enhancer hijacking"[17,21–24]. In addition, the overall distribution of somatic cancer mutations appears to be guided by 3D genome folding[25].

Recently, it became apparent that by mapping 3D genome organization using Hi-C (the whole-genome variant of the chromosome conformation capture (3 C) technology[26]), we can simultaneously obtain a highly-resolved map of SVs and CNVs genome-wide[27]. The emergence of tools like *hicbreakfinder* [28] and EagleC[29] allows for the systematic detection of SVs/CNVs in Hi-C data. Via such analyses, the functional impact of SVs on subtype-specific cancer gene regulation[19,30], as well as on a compendium of cancer lines has been investigated[24,28,31]. Still, our understanding of 3D genome organization in GBM remains limited due to the small number of samples analyzed to date (i.e., only 5 tumors in Ref.[27], and just 4 cell lines in refs. [32–34]) and to the absence of comprehensive SV/CNV co-analysis.

To address this and study the impact of patient-specific SVs, we derive glioblastoma stem cell-like cells (GSCs) from 24 *IDH*[wt] GBM patients—for three of which we could sample both the primary and the relapse tumor (see Supplementary Table 1). It is well acknowledged that the subset of GBM tumor cells with stem cell-like attributes are implicated in essentially all aspects of GBM initiation, maintenance, therapy resistance, recurrence, and tissue invasion in vivo[35,36]. Given that patient-derived GSCs retain the genomic and functional traits of their tumors of origin[37–39], they hold significant potential for translational modeling of GBM. Here, we generate 28 high-resolution Hi-C datasets and analyze them to map structural variation in each patient. We discover remarkable pervasiveness and variance in SV distribution across our cohort, giving rise to patient-specific "neo-TADs" and "neoloops". We then combine reassembled chromosomal scaffolds with matching transcriptome and histone mark data to understand how GBM gene expression and tumor recurrence are supported by such extensive heterogeneity in 3D genome folding.

## Results

### Pervasive and widespread structural variation along GSC chromosomes

We applied in situ Hi-C to 28 low passage *IDH*[wt] GSCs, including pairs derived from primary and recurrent tumors from 3 patients (Fig. 1a) to generate a total of ~19 billion read pairs. Following stringent filtering, we were left with >400 million valid read pairs per patient on average (63.4% mean data usage; Supplementary Table 2). This allowed us to produce 5 kbp resolution contact maps for each GSC, and confirm reproducibility by generating additional replicates from two randomly-selected lines (SCC > 0.93; Supplementary Fig. 1a).

We first addressed CNV prevalence in cancer cells[40] that can distort Hi-C contact maps. We verified that CNVs identified using whole-genome sequencing (WGS) data from an exemplary line, G181, were essentially identical to those computed via Hi-C data (Supplementary Fig. 1b) as was previously demonstrated[29,33]. Next, we applied CNV-based matrix-balancing[24] to Hi-C contact maps to alleviate any distortions that standard matrix balancing could not (Supplementary Fig. 1c) and CNV-balanced matrices were used for SV discovery in our cohort.

For a comprehensive identification of SVs in our GSC cohort, we applied *EagleC*[29] to 5 kbp resolution Hi-C matrices. SVs, even those with breakpoints separated by <100 kbp, produced characteristic signal aberrations in the contact matrices and were detected with high sensitivity (Supplementary Fig. 1d). In total, we mapped 2675 SVs across 28

Hi-C datasets on top of 591 complex SVs using EagleC (listed in Supplementary Data 1). These comprised 737 (27.6%) interchromosomal translocations, plus 713 (26.7%) intrachromosomal inversions, 652 (23.4%) deletions and 573 (21.4%) duplications. Of the 1938 intrachromosomal SVs, 57.6% were short- ( < 2 Mbp) and 42.4% long-range ( ≥ 2 Mbp) (for an example from G457 see Fig. 1b,c). Detection of SVs was robustly reproducible between replicates from the same line (mean Jaccard index = 0.63; Supplementary Fig. 1e). As a control, EagleC applied to astrocyte Hi-C data[33] returned only 7 SVs. A similar distribution of SVs was obtained when *hicbreakfinder* was used, returning a total of 1586 SVs (Supplementary Fig. 2a and Supplementary Data 1). Note that >53% of SVs returned by *hicbreakfinder* were also identified using EagleC. SV occurrence across our GSCs was pervasive, with 16 out of 28 samples carrying >80 SVs (the least number of SVs was 24 for G452C, and the most were 182 for G450). Notably, relapse tumor-derived GSCs usually carried more SVs than primary tumor-derived ones (Fig. 1d and Supplementary Fig. 2a). Taken together, these analyses demonstrate the sensitivity and reproducibility of SV discovery in our collection.

We next asked what the degree of SV recurrence is across our samples. One-to-one comparisons of HiC-deduced SVs from all samples (Supplementary Fig. 3) and similarity analysis (Fig. 1e and Supplementary Fig. 2b) revealed remarkable heterogeneity among GSCs and little recurrence (mean Jaccard index = 0.02). Even mutations known to be characteristic of GBM were only found in a subset of samples. For example, *EGFR* amplification[41] was associated with SVs in just 9 out of 28 lines, while *CDKN2A* deletion[42,43] was more prevalent and detected in 17 out of 28 lines. Finally, although SVs in GSC pairs from primary-relapse tumors of the same patient showed somewhat higher overlap (Jaccard index = 0.21), this did not increase much (Jaccard index = 0.42) once SVs from the central and peripheral part of the same tumor (i.e., G452C/P) were considered, further highlighting the high intra-patient heterogeneity of GBM.

Despite their uneven distribution across our collection, SV breakpoint positions correlated well with particular genomic features. For example, genomic duplications were strongly biased for strongly transcribed, GC-rich segments in the active chromatin A-compartment involving breakpoints close to TAD boundaries (using astrocyte Hi-C as reference; Fig. 1f). Translocations and inversions also involved gene-/GC-rich loci, but could be equally near or distal to TAD boundaries, agreeing with the notion that active gene co-association promotes rearrangements (especially in the case of translocations[44,45]). Conversely, deletions mostly occurred in AT-rich segments of the inactive B-compartment (Fig. 1f). Overall, we recorded significant enrichment for SVs in the A-compartment, particularly in gene-rich stretches and in the vicinity of TAD boundaries (Fig. 1f). This is in line with the preferential occurrence of DNA double-strand break hotspots within accessible, actively transcribed chromatin[46,47]. Notably, transcription start sites (TSSs) of genes linked to the GBM transcriptional program (derived from DisGenet[48]) were markedly enriched at breakpoint-associated TAD boundaries (Fig. 1g), suggesting that boundary disruption can favor oncogene dysregulation and malignant transformation[17,22,49]. This also held true for SVs, resulting in gene fusions. We identified 421 fusion events in mRNA-seq data generated from each GSC (Fig. 1a), but as Hi-C is more sensitive in detecting them[29], we could identify an additional 902 fusions using EagleC and 492 using *hicbreakfinder* (Supplementary Data 1) with 137 events found in both Hi-C and RNA-seq data. Gene fusions had significantly higher expression than their counterparts in astrocytes, but only when loci with a CNV ratio of >1.5 were considered (Supplementary Fig. 2c). This suggests that fusion gene overexpression is linked to genomic amplification in GBM.

Finally, rather than stochastically distributed along chromosomes, SVs show a propensity for clustering, especially in GC-/gene-rich segments (Fig. 1h and Supplementary Fig. 3b). Such high degree of

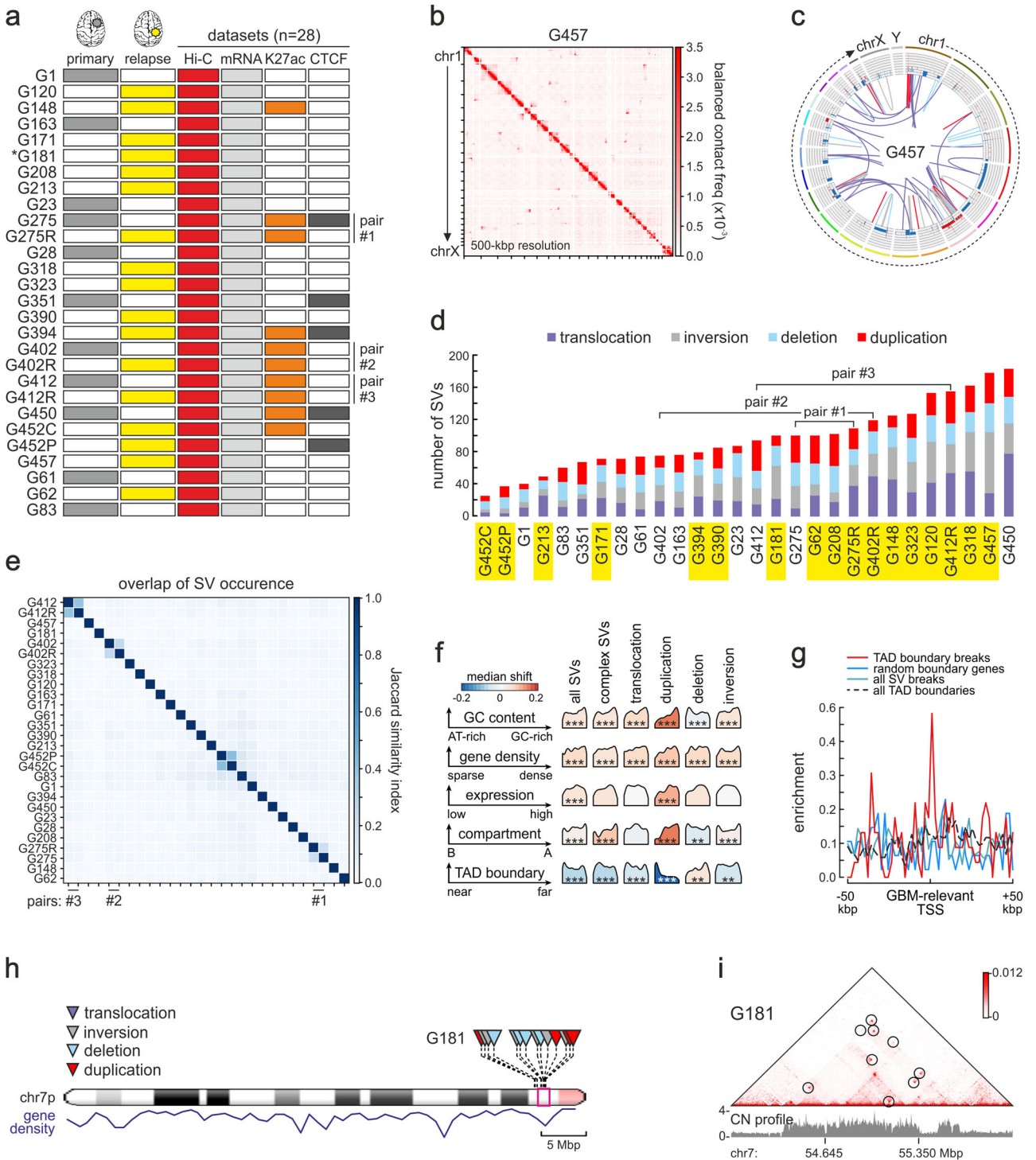

**Fig. 1 | Pervasive and uneven SV occurrence discovered by Hi-C analysis of patient-derived GSCs. a** Overview of our cohort from 28 primary (gray) or relapse GSCs (yellow). NGS data generated from each GSC are indicated. *: WGS data is available for G181; -R designates GSCs derived from the relapse tumor in a pair, and –C/P GSCs derived from the central or peripheral part of the same tumor. **b** Heatmap of 500 kbp-resolution Hi-C data along all G457 chromosomes. Strong interchromosomal signals represent translocations. **c** Circos plot of SVs and CNVs detected in G457 Hi-C using EagleC. Outer tracks: chromosomes; inner tracks: gain (red: >2 copies) or loss of genomic segments (blue: <2 copies); lines: translocations (purple), inversions (gray), deletions (light blue) or duplications (red). **d** Bar plot showing the number of SV types identified by EagleC in each GSC line. Lines from relapse tumors are highlighted (yellow). **e** Jaccard similarity index of SVs discovered in different Hi-C datasets. **f** Enrichment of breaks from all SV types (columns)

relative to GC content, gene density, gene expression, A/B compartment, or TAD boundaries (rows). Each density curve represents the quantile distribution of the particular genomic feature at SV breakpoints compared to random positions. **: FDR < $10^{-3}$; ***: FDR < $10^{-5}$ calculated after multiple hypothesis corrections on a one-sided Kolmogorov-Smirnov test based on a sample size of 5,078 genomes containing SVs. **g** Mean enrichment of GBM-associated gene TSSs in the 100 kbp around TAD boundaries from astrocytes with an SV break in GBM (red), all SV breaks (green), randomly-selected TAD boundary-associated genes (blue) or all TAD boundaries (dashed black). **h** Ideogram of chr7 showing SV distribution from G181 Hi-C data (above) and gene density (below). **i** Exemplary Hi-C contact map from G181 in a 2-Mbp region of chr7 (magenta in panel h) harboring multiple SVs (circles).

breakpoint clustering (almost 43% of EagleC-deduced SVs, i.e., 2298 out of 5350, were in clusters) led to complex rearrangements within relatively small (<2 Mbp) genomic stretches (Fig. 1i and Supplementary Fig. 3b, c). Notably, smaller chromosomes like chr12 (also remarked on in TCGA WGS data analysis[41]), chr16 and chr17 carried a disproportionately high density of SVs (i.e., >3.5 SVs/Mbp compared to a median of <2 SVs/Mbp; Supplementary Fig. 3c). These results highlight structural variation in GSCs as a highly pervasive source of heterogeneity bound to change the 3D regulatory architecture of GBM chromatin.

## GSC-specific chromatin organization blurs transcriptional subtype classification

Despite GBM heterogeneity, bulk and single-cell transcriptome analyzes have been used to classify GBM entities into few major subtypes[50–53]. Wang and colleagues[51] classified GBMs as classical (TCGA-CL), proneural (TCGA-PN), or mesenchymal (TCGA-MES). Neftel and colleagues[53] combined single-cell analysis of tumors and TCGA specimens to define four main GBM states – mesenchymal (MES-like), astrocyte-like (AC-like), oligodendrocyte-progenitor-like (OPC-like), and neural progenitor-like (NPC-like) – which reflect distinct neural cell types influenced by their microenvironment. Finally, Richards and colleagues[53] analyzed a collection of GSCs to uncover a transcriptional gradient spanning two cellular states reminiscent of neural development (DEV) and inflammatory injury response (INJUR).

We used our mRNA-seq data from our 28 GSCs (Fig. 1a) in conjunction with each of these classifications in order to subtype them. In PCA plots, GSCs separated well from astrocytes and generated a gradient amongst themselves, in which GSCs from the same patient (i.e., primary-relapse or central-peripheral pairs) separated the least, suggesting that intra-patient gene expression differences are less than inter-patient ones (Supplementary Fig. 4a). We next applied single-sample gene set enrichment analysis (ssGSEA) using the three classifications discussed above[51–53] to find that most of our GSCs as significantly associated (empirical $P$-value < 0.01) with a given subtype (Supplementary Fig. 4a, b). In all three cases, we found 5-6 GSCs that did not enrich for any of the provided signatures and, especially for the 4-cell state classification[52], many samples with ambiguous, mixed subtyping (Supplementary Fig. 4b). Still, the ME/PN/CL and DEV/INJUR classifications provided reasonable subclustering of gene expression profiles for most of our GSCs (Supplementary Fig. 4a). Reassuringly, the mesenchymal-like GSCs in two different classifications overlapped for their most part (Supplementary Fig. 4a, b).

Finally, as subtypes of other cancers (e.g., acute myeloid leukemia) were recently shown to classify on the basis of large-scale (i.e., compartmental) 3D genome organization features[31], we asked whether different hierarchical features in our Hi-C data also allow classification of GSCs into the subtypes deduced above. To this end, we first removed SV-/CNV-affected regions from our data (which should explain most inter-patient divergence; see Fig. 1e) and then used different Hi-C features like compartments (using the first principal component, PC1, of 50 kbp resolution eigenvectors), insulation scores delineating TAD boundaries (computed at 25 kbp resolution), all Hi-C contacts (computed at 10 kbp resolution) or loops (at 5 kbp resolution) for data clustering. We found that differential PC1 calling, reflecting GSC-specific differences in eu-/heterochromatin, broadly separated many (but not all) CL/DEV from MES/INJUR lines, yet this was not a reflection of the presence or absence of major GBM driver mutation like amplification of *EGFR* or loss of *CDKN2A/B* and *PTEN* (Supplementary Fig. 4c). All higher-resolution features discriminated only moderately (i.e., insulation score/Hi-C contacts) or very little (i.e., loops) between subtypes (Supplementary Fig. 4d–f). Even samples with highly similar gene expression profiles like the MES/INJUR G83 and G457 or the CL/DEV G120 and G412R lines (Supplementary Fig. 4a) did not cluster together (Supplementary Fig. 4c–f). We attribute this to

the pervasive structural heterogeneity that underlies individuality of each patient-derived line.

## GSC-specific SVs underlie neo-domain and neo-loop formation

Induction of SVs along chromosomes does not simply disturb the integrity of chromosomes and the continuity of gene loci, but also reorganizes 3D spatial interactions of chromatin to give rise to new topological domains, termed "neoTADs"[28,54]. We mapped their formation across all 28 Hi-Cs to identify a total of 2222 neoTADs using EagleC and 1401 using *hicbreakfinder* (with 740 found by both tools). They had a median size of 0.5 Mbp and arose from all SV types (Supplementary Fig. 5a, b). Different GSCs carried vastly different numbers of neoTADs (from 10 in G452C to 244 in G450; Supplementary Fig. 5b), again highlighting a remarkable heterogeneity in these GBM specimens. Notably, expression of genes within neoTADs was consistently higher than that of genes in neighboring TADs (Supplementary Fig. 5c), as well as of the same genes in astrocytes or GSCs where the specific neoTAD did not form (Supplementary Fig. 5d). This potentiation of gene expression holds true even when loci with a CNV-ratio >1.5 were filtered out (Supplementary Fig. 5c, d). Therefore, neoTADs support GSC-specific gene activity.

Similarly to neoTADs, SVs also gave rise to "neoloops" characteristic of the patient-derived lines (Fig. 2a). We used *Neoloopfinder*[24] to identify neoloops in locally reconstructed and normalized for allelic effects Hi-C maps from all 28 GSCs. Using an FDR cutoff of <0.05, we found 6331 neoloops across EagleC-deduced SVs and 5297 across *hicbreakfinder*-deduced ones (with 2115 shared by both; Supplementary Data 1). Again, the number of neoloops in each GSC varied significantly (from 12 in G28 to 1327 in G148, with a median of 120; Fig. 2b and Supplementary Fig. 6a), but we saw little correlation between the number of SVs and neoloops in our collection ($\rho = 0.29$) meaning that, depending on the cellular context, even the same SVs will not always give rise to neoloops. Hence the need for Hi-C data incorporation on top of one-dimensional epigenomic and WGS data.

Approximately 50% of EagleC-deduced neoloops in GSCs for which we also have CTCF binding information (Fig. 1a) were anchored at CTCF-bound sites (i.e., 772 out of 1579; Fig. 2c), with 88.5% of them abiding to the expected convergent CTCF motif orientation[55]. More than 90% of neoloops were <0.8 Mbp in size (Fig. 2d), and we identified 2053 genes associated with neoloops (i.e., within ±5 kbp of either anchor; more than expected by chance, $P = 0.017$ Fischer's exact test). Of these, 858 were protein-coding, and 131 of these protein-coding genes recurrently associated with neoloops in our cohort, albeit at a low mean recurrence of 2. Amongst them, 33 (25.2%) have been reported as GBM-related (e.g., *EGFR*, *PTEN*, *MTOR*) and 29 (22.1%) as cancer-associated genes (e.g., *AGAP2*, *SOX2*). A query using all these neoloop-associated genes in DisGeNET[48] returned a strong enrichment for characteristic glioma and GBM programs (Fig. 2e), including genes with a high disease specificity index, like *SYF2* or *AGAP2*. Notably, neoloops sustained significantly higher expression of their associated genes in a GSC-specific manner (Fig. 2f–h; with a cutoff of $\log_2(\text{TPM}+1) > 0$), which contributes to inter-patient transcriptional heterogeneity. These observation also held true when *hicbreakfinder*-deduced neoloops were used for the analysis (Supplementary Fig. 6b, c).

Much like SV profiles that were largely idiosyncratic, neoloop recurrence among GSCs was also limited. For instance, the most correlated at the loop level unpaired samples, G1 and G213 (Supplementary Fig. 4f), shared <10% of their neoloops, while even the two intratumor-derived lines G452C/P shared <42%. Nevertheless, even limited recurrence becomes relevant in cases where neoloops associated with a particular gene locus in different GSCs. In total, 858 protein-coding (plus 1195 non-coding) genes associated with neoloops in our collection. Of these, 40 protein-coding (plus 59 non-coding) genes were neoloop-associated in 3 or more GSCs. For example,

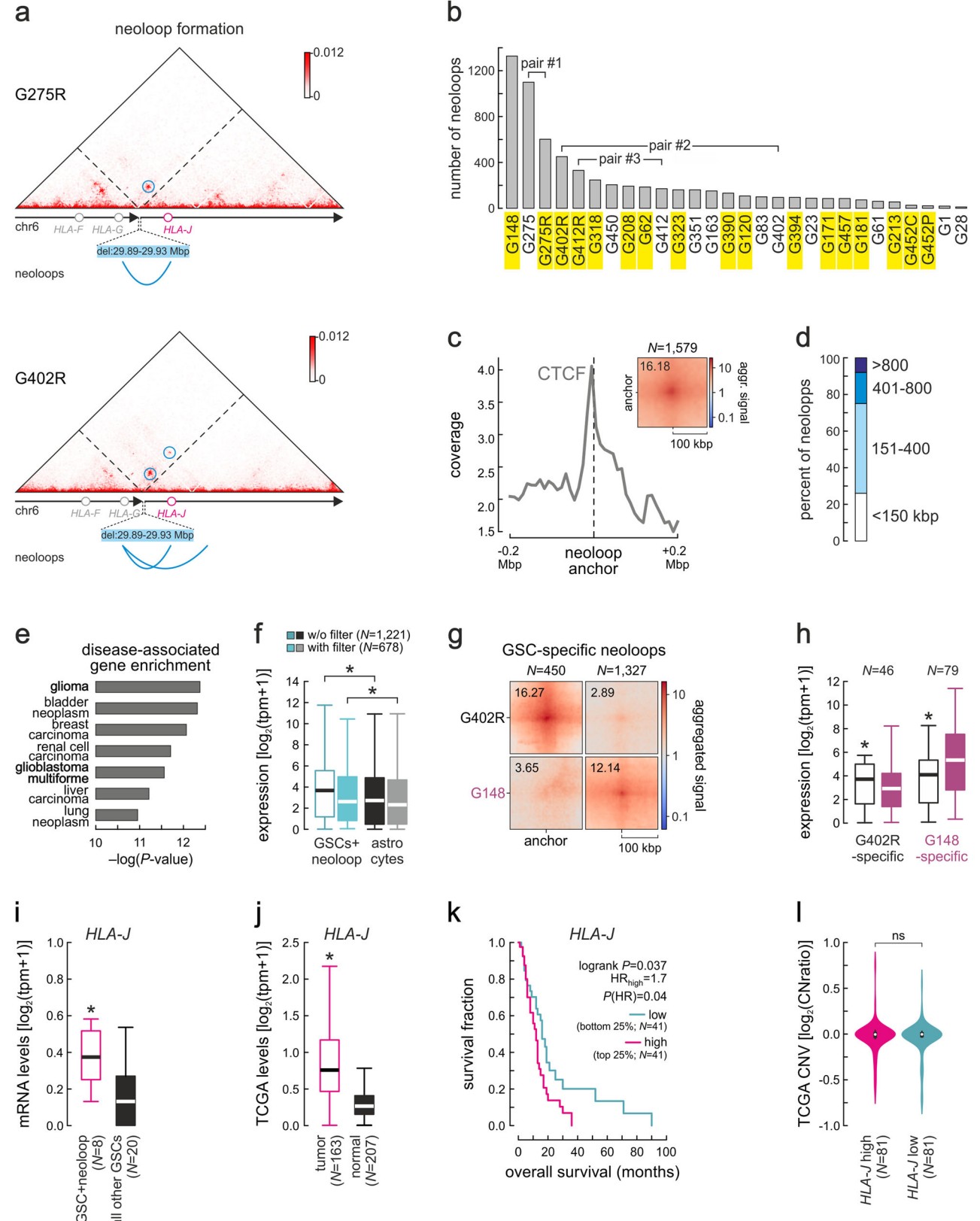

neoloops formed in the *HLA-F/-G/-J* locus following a 50-kbp deletion in 8 lines. These neoloops encompassed *HLA-J* (Fig. 2a), which has been associated with poorer prognosis in melanoma[56] and breast cancer patients[57]. *HLA-J* was expressed significantly higher in GSCs carrying neoloops in its locus compared to all other in our cohort (Fig. 2i). It was

also significantly higher expressed in TCGA data from GBM tumors compared to control tissue (Fig. 2j). Importantly, *HLA-J* overexpression associates with poorer GBM patient survival overall, but this is not due to gene amplification (Fig. 2k, l). Another example are the neoloops forming in the *ADAM9* locus (Supplementary Fig. 6d) encoding a cell-

**Fig. 2 | Extensive neoloop occurrence supports GSC-specific programs.**
**a** Exemplary Hi-C contact maps from G275R and G402R around a 50 kbp deletion in the *HLA* locus. Neoloops forming across the breakpoint are indicated (blue circles). **b** Bar plot showing the number of neoloops identified in each GSC line based on EagleC SVs. Lines derived from relapse tumors are indicated (yellow). **c** Line plot showing CTCF binding signal in the 400 kbp around all neoloop anchors. Inset: Aggregate peak analysis (APA) plot for all neoloops detected. **d** Bar plot showing the percent of neoloops of different sizes. **e** Signatures of neoloop-associated genes from the DisGeNET database; *P*-values calculated using two-sided Fisher's exact tests without multiple comparison adjustments. **f** Box plots (band shows the mean, each box extends between 1ˢᵗ and 3ʳᵈ quartile, and whiskers extend 1.5x the interquartile range) showing mean expression of neoloop-associated genes in GSCs with neoloops (green) or astrocytes (black) with and without filtering out loci with >1.5 CNV. *: $P = 1.377e$-51 (left) and $2.435e$-09 (right), two-sided Wilcoxon rank-sum test. Source data for this panel are provided as a Source Data file. **g** APA plots

of neoloops specific to G402R or G148. **h** Box plots (drawn as in panel f) show expression of genes associated with GSC-specific neoloops from panel f. *: $P = 3.472e$-03 (left) and $8.179e$-07 (right), two-sided Wilcoxon rank-sum test. Source data for this panel are provided as a Source Data file. **i** Box plots (drawn as in panel f) show *HLA-J* expression in GSCs carrying (magenta) or not neoloops in this locus (black). *$P = 0.008$; two-sided Student's t-test. Source data for this panel are provided as a Source Data file. **j** Box plots (drawn as in panel f) show *HLA-J* expression in TCGA GBM tumor and matching normal tissue *$P < 0.01$; two-sided Mann-Whitney U-test. **k** Kaplan-Meier survival curves for GBM patients with *HLA-J* high/low expression. *P*-value calculated using a two-sided log-rank test. **l** Violin plots (medians indicated by white circles, and 1ˢᵗ/3ʳᵈ quartile span by black boxes) showing CNV in the *HLA-J* locus from TCGA GBM tumors with high (top 25%, magenta) or low *HLA-J* expression (bottom 25%, green). $P = 0.184$; two-sided Mann-Whitney U-test. Source data for this panel are provided as a Source Data file.

---

surface protease implicated in solid tumor biology[58]. *ADAM9* is over-expressed in neoloop-carrying GSCs in our cohort as well as in TCGA GBM tumors (Supplementary Fig. 6e, f). This not due to gene amplification and predicts poorer patient disease-free survival (Supplementary Fig. 6g, h). Together, this data suggests that neoloop formation can be instructive of patient-specific gene expression patterns, thereby helping us uncover hitherto unknown GBM dependencies and prognostic markers.

## Enhancer-promoter neoloops explain GSC-specific gene dysregulation

Given that expression of genes was higher in GSCs in which they associate with neoloops (Supplementary Fig. 6a, b), we wanted to further investigate how neoloops contribute to gene dysregulation in GBM. To this end, we used Hi-C data from 10 GSCs for which we also generated H3K27ac CUT&Tag profiles (Fig. 1a). Using these histone mark profiles on locally-reassembled genomic scaffolds, we were able to identify putative active enhancers in each line and see that many of our EagleC-identified neoloops actually represented enhancer-promoter (E-P) contacts driving GSC-specific gene expression (see the example of *IFIT1* in Fig. 3a, b).

Of 4343 neoloops and 11031 non-neoloops (i.e., loops within a local assembly that do not span an SV breakpoint) in these 10 GSCs, E-P neoloops were significantly overrepresented (35.3%) compared to non-neoloops (25.5%; $P < 10^{-5}$ Fisher's exact test), while the converse applied to promoter-promoter (P-P; 9.2% neo- vs 16.9% non-neoloops) and other loops (33.3% neo- vs 66.6% non-neoloops; Fig. 3c). Critically, E-P neoloops linked putative enhancers to 192 gene TSSs in these GSCs to induce expression levels higher than those of 801 genes linked to non-neoloop enhancers. This also held true for P-P neoloops and their 243 associated genes versus 1055 non-neoloop-associated ones (Fig. 3d). Similarly, the levels of 238 E-P neoloop-associated genes (85 when amplified loci were filtered out) were significantly higher in these 10 GSCs compared to lines where neoloops do not form (Fig. 3e). Gene ontology (GO) analysis of E-P/P-P-associated genes revealed that they were linked to key cancer-related pathways like "GBM signaling", "mitotic cell cycle", "senescence" or "chromatin organization" (Fig. 3f) arguing in favor of a tumor-specific importance of E-P neoloops.

Finally, we compiled a list of 138 known GBM-/cancer-related genes that we could link with a neoloop and with at least one SV-linked event (i.e., with a deletion, duplication or fusion) in our collection. Of these, 43 associated with at least two such events across our samples, and we saw that dysregulation could be attributed almost just as often to neoloop formation in the locus as to the duplication, deletion or fusion of the gene itself (Fig. 3g). These observations suggest that E-P neoloops are a regulatory hallmark of GBM, and highly selected for in the course of tumor development to sustain patient-specific gene expression patterns.

## Modeling neoloop formation as a selective GBM dependency

The pervasive and uneven emergence of SVs and neoloops in our cohort would inevitably give rise to GSC-specific E-P interactions and ensuing gene activation. This creates an opportunity of potential translational value if we could identify tumor-specific dependencies arising from E-P neoloops activating druggable gene targets or pathways. To exemplify this, we selected G148 in which *MYC* is markedly overexpressed. This was due to a focal co-amplification of the *MYC* gene locus on chr8 together with an Mbp-long segment on chr12 via a translocation (Supplementary Fig. 7a). This SV brings into spatial vicinity *MYC* with a cluster of co-amplified enhancers via the formation of neoloops (Fig. 4a) in a structure likely representing extra-chromosomal DNA (see putative breakpoints mapped in Supplementary Fig. 7a). This combination of enhancer hijacking and co-amplification resulted in >10-fold increase in *MYC* mRNA levels in this line compared to all other GSCs (or astrocytes; Fig. 4b), which was also reflected on MYC protein levels (Supplementary Fig. 8a).

We therefore tested whether targeting MYC would selectively inhibit G148 growth. We treated G148, as well as G62 cells, where no enhancer hijacking, amplification or *MYC* overexpression occurs (Fig. 4b), with the small molecule inhibitor, EN-4. EN-4 specifically targets Cys171 of MYC to form a covalent bond and impair its binding to target genes[59]. GSC treatment with EN-4 led to the selective suppression of MYC and cell proliferation (marked by Ki-67 expression) in *MYC*-overexpressing G148, but not in G62 (Fig. 4c and Supplementary Fig. 8b,c). Moreover, EN-4 treatment led to a significant increase in DNA damage and cell death in G148 compared to G62, as deduced from TUNEL assays (Fig. 4d and Supplementary Fig. 8d).

Given this selective dependency of G148 on *MYC* overexpression, we argued that being able to predict the formation of gene-activating E-P neoloops on a patient-specific basis could inform treatment options, as such neoloops are selected during tumor development and could represent new dependencies. To achieve this, we built on the in silico PRISMR approach previously used to predict ectopic regulatory interactions due to congenital disease-causing SVs[60]. In its original implementation, PRISMR could only predict such interactions in *cis* by inferring binding site distribution along the polymer that best reproduces the Hi-C matrix of a genomic region in its wild-type configuration. Then, ectopic interactions are predicted by reshuffling the polymer in accordance with SVs and recalculating new Hi-C contacts[60]. As we wanted to model a structural variant occurring in *trans*, we modified the approach to infer binding site distribution in an extended segment of chr12 (i.e., chr12: 57.66-58.33 Mbp) using data from G275R that does not carry any SVs in the region, but in conjunction with RNA-seq and H3K27ac signal from G148 to ensure faithful Hi-C contact prediction using a probabilistic approach (see Supplementary Fig. 7b and "Methods" for details). Following optimization of binding classes (Supplementary Fig. 7c,d), we found that the three best-correlated

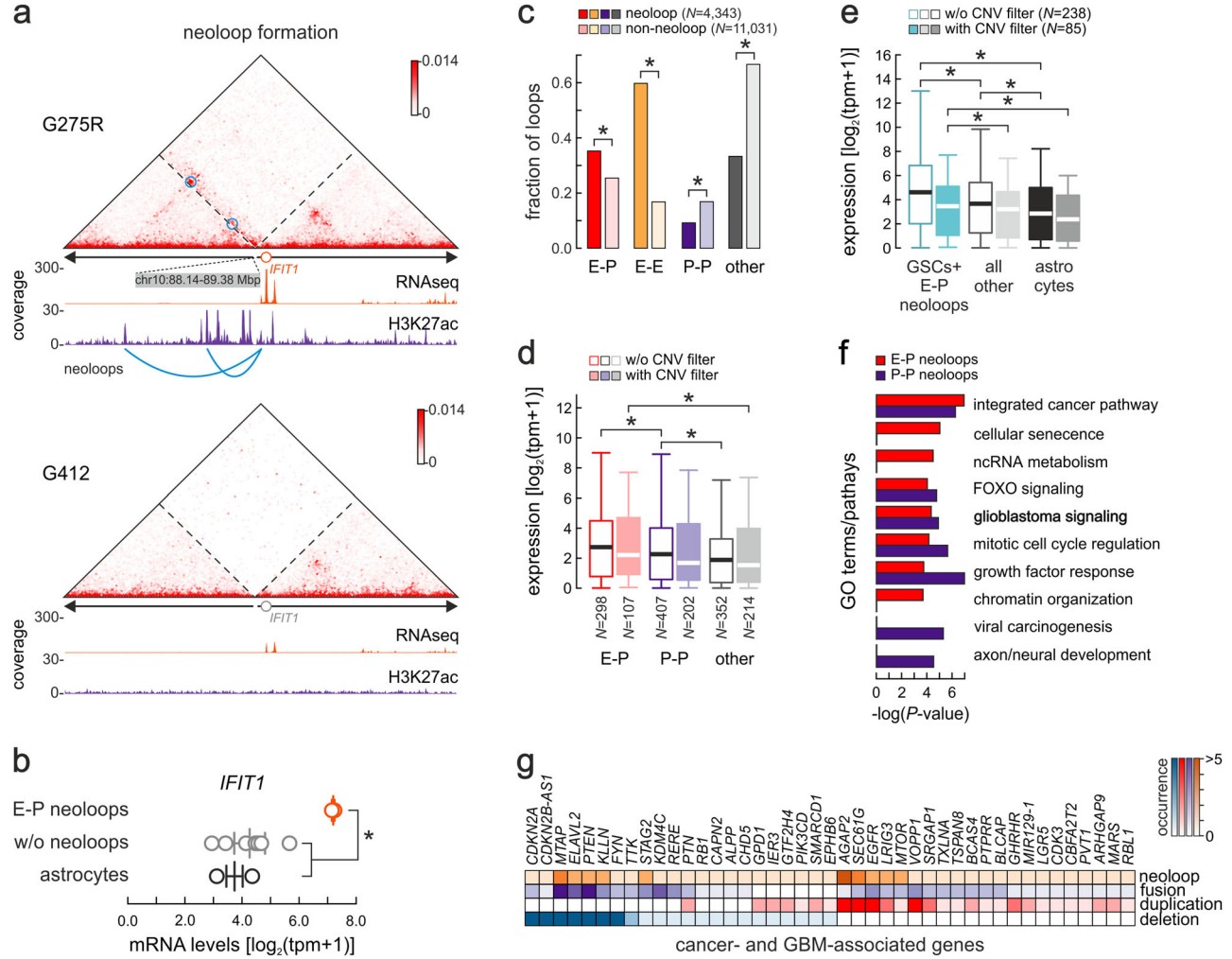

**Fig. 3 | Enhancer-promoter neoloops in GSC-specific gene regulation.**
**a** Exemplary Hi-C contact map from G275R around a 1.24-Mbp inversion in the *IFIT1* locus. Enhancer-promoter neoloops forming across the breakpoint are indicated (blue circles). The same locus of G412, where no inversion occurs, provides a control. **b** Plot showing mean (line) ±SEM and GSC-specific *IFIT1* RNA-seq levels (circles) in lines with (orange; n = 2) or without the inversion (gray; n = 8) or in astrocytes (black; n = 2). *: P = 0.0042, unpaired two-sided Student's t-test. Source data for this panel are provided as a Source Data file. **c** Bar plot showing the fraction of neo- or non-neoloops around EagleC SVs representing enhancer-promoter (E-P), enhancer-enhancer (E-E), promoter-promoter (P-P) or other loops. *: P = 6.34e-34 (E-P), 0.1e-99 (E-E), 9.08e-34 (P-P) and 0.1e-99 (other), two-sided Fisher's exact test. Source data for this panel are provided as a Source Data file. **d** Box plots (bands show the mean, each box extends between 1st and 3rd quartile, and whiskers extend

1.5x the interquartile range) comparing expression of genes involved in enhancer-promoter (E-P), promoter-promoter (P-P) or other neoloops with and without filtering out loci with >1.5 CNV. *: P = (left to right) 0.0007, 0.0447 and 0.0053, two-sided Mann-Whitney U-test. Source data for this panel are provided as a Source Data file. **e** As in panel (**d**) but showing expression of E-P neoloop-associated genes in GSCs with (green) or without neoloops (gray) or in astrocytes (black). *: P = (left to right) 4.55e-20, 4.47e-20, 0.023, 0.039, and 1.82e-10, two-sided Wilcoxon rank-sum test. Source data for this panel are provided as a Source Data file. **f** GO terms associated with genes involved in E-P (red) or P-P neoloops (blue); P-values calculated using two-sided Fisher's exact tests without multiple comparison adjustments. **g** Heatmap showing duplication, deletion, neoloop, or gene fusion occurrence for 43 known cancer-associated gene loci.

ones in G148 recapitulated input RNA-seq and H3K27ac profiles (Fig. 4e and Supplementary Fig. 7b).

In turn, the inferred classes of binding sites gave rise to highly similar contact matrices for the experiment and simulation (ρ = 0.65). In G148, we could predict the formation of neoloops connecting *MYC* on chr8 to the active putative enhancers on chr12 (Fig. 4f, left), while for G394, where no translocation and amplification occur, no neoloops were predicted (Fig. 4f, right). This was also reflected in the 3D models rendered from the simulations, whereby hubs between hijacked enhancers and the *MYC* locus formed in G148 (Fig. 4g) but not in G394 data (Supplementary Fig. 7e). Remarkably, our 3D models showed the mutually exclusive formation of contacts between the *MYC* promoter and enhancer beads of either class 1 or 7 (Fig. 4e–g and Supplementary Fig. 7f). These differential conformations (Fig. 4g) gave rise to a

heterogeneous population of 3D models, which reflected different predicted *MYC* activation levels (calculated as in ref. 61). Models with different enhancer-promoter contacts cluster away from one another and lead to variable levels of *MYC* activation (Supplementary Fig. 7g, h). Thus, we can now model the impact of SVs also in *trans* using a minimal set of epigenetic tracks to model structural aberrations also in *trans* that could represent potential patient-specific vulnerabilities.

### GBM relapse associates with divergent SV occurrence and 3D genome folding
GBM is of the most aggressively recurring tumors, with patients usually succumbing within ~1 year of relapse[1]. Our collection included GSCs from three such primary and relapse tumor pairs (G275/R, G402/R, and G412/R; Fig. 1a), and we generated Hi-C, mRNA-seq, and H3K27ac data

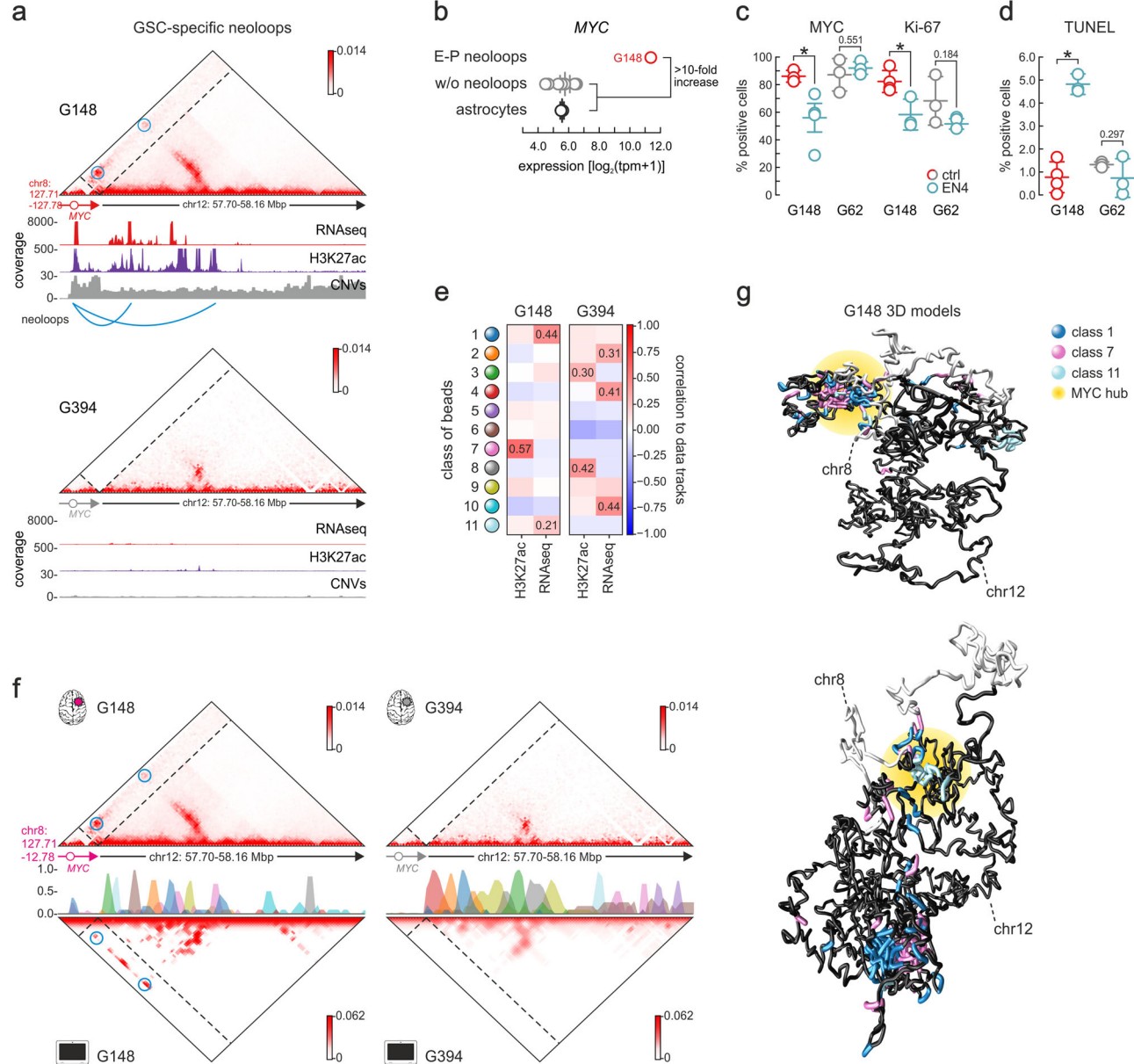

**Fig. 4 | GSC-specific dependencies uncovered by neoloop analyzes and simulations. a** Exemplary Hi-C contact map from G148 around a translocation breakpoint involving the *MYC* locus. Enhancer-promoter neoloops forming across the breakpoint are indicated (blue circles) and aligned to RNA-seq, H3K27ac, and CNV tracks. Absence of neoloops in G394 (below), where no translocation occurs, provides a control. **b** Plot showing mean (line) ±SEM GSC-specific *MYC* expression (circles) in cells with (orange; *n* = 1) or without the chr8:chr12 translocation (gray; *n* = 9) or in astrocytes (black; *n* = 2). Source data for this panel are provided as a Source Data file.**c**, Plots showing the percentage of cells ±SD staining positive for MYC or Ki-67 in untreated (red) or EN4-treated *MYC*-high G148 (green) from *n* = 4 independent experiments; treatment of the *MYC*-low G62 provides a control. *: *P* = 0.038 (for MYC) and 0.021 (for Ki-67), unpaired two-sided Student's t-test. Source data for this panel are provided as a Source Data file. **d** As in panel (**c**) but showing the percentage of cells ±SD positive for TUNEL staining from *n* = 4

independent experiments. *: *P* = 0.00025, unpaired two-sided Student's t-test. Source data for this panel are provided as a Source Data file. **e** Heatmaps showing correlation of each class of polymer beads with H3K27ac and RNA-seq data from G148 that carries the chr8:chr12 translocation (left) or G394 that does not (right). Classes with a correlation of >0.2 are shown. **f** Left: Contact maps from Hi-C (top) or simulations (bottom) around the translocation breakpoint in G148 are shown aligned to polymer bead classes. Enhancer-promoter neoloops forming across the breakpoint are indicated (blue circles). Right: As in the left hand-side panel, but for G394 that does not carry the translocation. **g** Representative 3D renderings of the two major configurations resulting from chr8:chr12 ecDNA translocation involving the *MYC* locus. Beads from binding classes 1, 7, and 11 that best predict folding are color-coded as in panel **d**, and differential *MYC*-enhancer interactions indicated (yellow halo).

from these pairs in an attempt to understand their reemergence following therapy keeping in mind that the standard-of-care for GBM patients is highly mutagenic radiochemotherapy.

A first observation was that, like all other GSCs we studied, these paired lines also showed high divergence in the number (Fig. 1d) and position of SVs mapped after applying either EagleC or *hicbreakfinder* to Hi-C (Fig. 5a and Supplementary Fig. 9a, b).

Relapse lines G275R, G402R, and G412R shared 33%, 33%, and 47% of EagleC-deduced SVs with their primary tumor-derived GSCs, respectively (with overlap being quite similar for *hicbreakfinder*-deduced SVs; Fig. 5b and Supplementary Fig. 9c, d). Gene expression upon relapse also diverged markedly (with just 30 up- and 8 downregulated genes shared by all three pairs, and <14% shared by any two pairs; Supplementary Fig. 9e). They also displayed, in

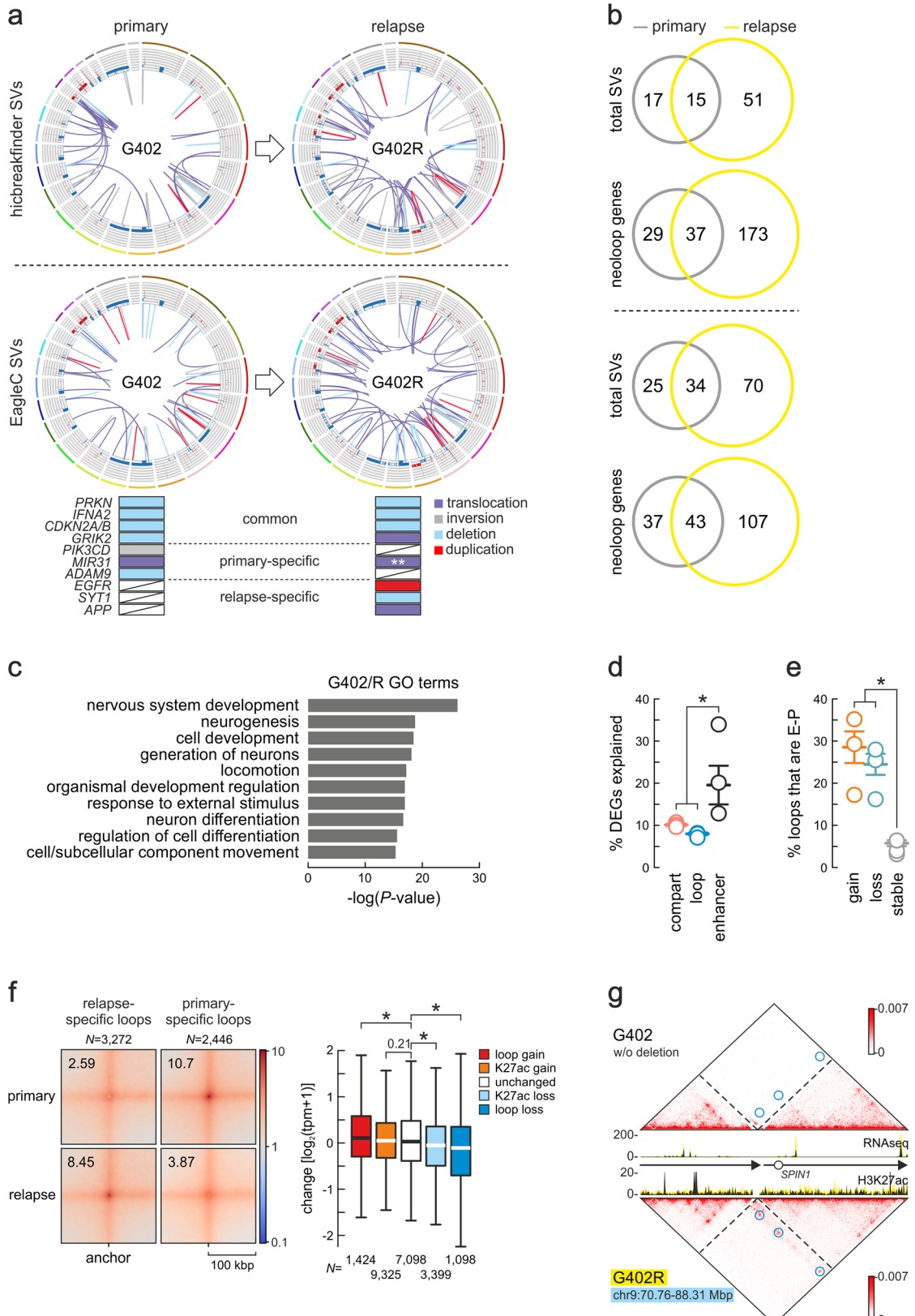

pairwise comparisons, the accumulation of new SVs/CNVs in known GBM and general oncogenic drivers (e.g., *EGFR* or *CDK6* amplifications in G402R and G275R, respectively; Fig. 5a and Supplementary Fig. 9a, b). Interestingly, when all pairs were considered, relapse GSCs showed a statistically significant increase in translocations (from 40 in primary lines to 90 in relapse ones – $P < 0.001$, Fisher's

exact test) associated with GBM-relevant genes (gda>0.01, listed in Supplementary Data 2), while deletions and inversions decreased by ~12% ($P < 0.05$, Fisher's exact test). In spite of all this, differentially expressed genes did converge to a few pathways in relapse-derived GSCs, like "neurogenesis" or "nervous system development" (Fig. 5c and Supplementary Fig. 9f).

**Fig. 5 | 3D genome folding differentiates relapse from primary GBM samples.**
**a** Circos plots of SVs and CNVs in G402 and G402R identified by *hicbreakfinder* (top) or EagleC (bottom). Outer tracks: chromosomes; inner tracks: gain (red: >2 copies) or loss of genomic segments (blue: <2 copies); lines: translocations (purple), inversions (gray), deletions (light blue) or duplications (red). Aligned below the Circos plots are the top GBM-associated genes (gda>0.01) that are common to both lines or specific to each, and linked to a particular SV type (color-coded).
**\*\***: The *MIR31*-associated translocation in the relapse line is different to the primary one. **b** Venn diagrams showing shared and unique SVs or neoloop-associated genes in primary and relapse Hi-C data identified by *hicbreakfinder* (top) or EagleC (bottom). **c** GO terms associated with differentially-expressed genes in primary versus relapse G402. *P*-values calculated using two-sided Fisher's exact tests without multiple comparison adjustment. **d** Percent (mean ± SEM) of differentially-expressed genes explained from n = 3 primary/relapse pairs by A/B-compartment, loop or enhancer changes in all GSC pairs. \*: *P* = 0.0145, unpaired two-sided Student's t-test. Source data for this panel are provided as a Source Data file. **e** As in panel d, but for differentially-expressed genes linked to E-P loops gained (orange), lost (green) or not changed upon relapse (gray). \*: *P* = 0.0331, unpaired two-sided Student's t-test. Source data for this panel are provided as a Source Data file. **f** Left: APA plot for neoloops specific to primary or relapse GSC pairs. Right: Box plots (bands show the mean, each box extends between 1st and 3rd quartile, and whiskers extend 1.5x the interquartile range) showing changes in the expression of genes associated with loops gained (red) or lost (blue), having increased (orange) or decreased H3K27ac (light blue), or not changing in relapse (white). \*: *P* (from left to right) = 1.60e-05, 2.09e-01, 3.26e-12 and 4.26e-12, two-sided Wilcoxon-Mann-Whitney test. Source data for this panel are provided as a Source Data file. **g** Hi-C contact maps around a 17.6-Mbp deletion on chr9 specific to G402R shown aligned to overlaid RNA-seq and H3K27ac tracks from primary (black) and relapse samples (yellow). G402-specific neoloops are indicated (blue circles).

We next asked which level of 3D genome organization was most involved in the divergent transcriptional profiles we recorded. Switching between A- and B-compartments or association with chromatin loops each explained 10% or less of the gene expression changes seen (Fig. 5d). Association with enhancers though explained on average twice as many differentially-expressed genes (Fig. 5d), and a larger fraction of E-P loops was dynamically lost or gained between the GSC pairs, thus highlighting their regulatory significance (Fig. 5e). We identified ~800 primary- and ~1000 relapse-specific loops per pair (826 and 327 for G275/R, 681 and 1366 for G402/R, and 934 and 1531 for G412/R; Fig. 5f and Supplementary Fig. 9g,h). We stratified these loops on whether they were specific to primary ("lost") or relapse GSCs ("gain"), shared by a GSC pair but showing increase ("K27ac gain") or decrease in H3K27ac signal in relapse ("K27ac loss") or remained unchanged. After assigning genes to the promoter anchor of these loops, we found that expression levels of thousands of genes from all three pairs were on average significantly higher (for "gain" loops) or lower (for "loss" loops) than those of genes associated with unchanged loops (Fig. 5f and Supplementary Fig. 9g, h). A considerable fraction of these loops were neoloops forming as a result of primary- or relapse-specific SVs. These neoloops often facilitated enhancer hijacking leading to aberrant gene overexpression (see example in Fig. 5g). Remarkably though, we found little recurrence of misexpressed genes associated with neoloops or with other SV types within a pair. Specifically, G275R, G402R and G412R shared 59%, 29%, and 34% of neoloop-associated genes with their primary tumor-derived counterparts, respectively (with overlap being similar for neoloops identified based on *hicbreakfinder*-deduced SVs; Fig. 5b and Supplementary Fig. 9c, d). This suggests that GBM relapse, as reflected in GSCs, associates with a set of SVs that are not a mere evolution of those in the primary tumor. Moreover, despite some convergence in the pathways affected, we saw marked individuality in the transcriptional programs of each GSC pair supported by 3D genome rewiring.

Finally, we used Hi-C and RNA-seq data to compare a pair of GSCs derived from the central and peripheral regions of the same GBM tumor (i.e., G452C/P). Surprisingly, we found that the periphery only shared ~50% of its SVs with the center regardless of EagleC or *hicbreakfinder* being used, with multiple G452C SVs lost in G452P (Supplementary Fig. 10a, b). This held true also for the genes associated with neoloops in the pair (Supplementary Fig. 10b). In line with the transcriptional subtype divergence of these GSCs (Supplementary Fig. 4a, b), we found pathways like "nervous system development", "regulation of signaling" or "ECM organization" enriched in the central over the peripheral GSCs (Supplementary Fig. 10c) in conjunction with ~200 prominent GSC-specific loops in either line (Supplementary Fig. 10d). This further affirms the pervasive heterogeneity in our GBM-derived collection, to the extent that even different parts of the same tumor diversify at the level of 3D genome architecture and gene regulation.

## Discussion

We generated 5 kbp-resolution Hi-C maps from 28 patient-derived GSCs and used their contact structure to identify tens to hundreds of SVs per sample. This highly resolved view of genomic rearrangements revealed a pervasive (16 out of 28 samples carried >80 SVs) yet very uneven SV distribution among samples (even between GSCs derived from two different parts of the same tumor). Despite their extreme heterogeneity and largely non-recurrent nature, SVs were not stochastically distributed along chromosomes. They clustered together in hotspots aligning well with GC-/gene-rich regions in the A- (i.e., transcriptionally active) chromatin compartment and near TAD boundaries. When we focused on SV breakpoints near TSSs of genes associated with the GBM transcriptional program, we found enrichment for TAD boundaries. This affirms that disruption of positions of 3D chromatin insulation favors oncogene activation, malignant transformation, and tumor growth[62]. Notably, in gliomas with *IDH* gain-of-function mutations, hypermethylation of CTCF sites at insulator elements prevents binding to also disrupt boundary formation[22]. Thus, we can envisage the development of interventions that act to preserve TAD boundary integrity and counteract GBM progression in the future.

However, our collection consists exclusively of *IDH*^wt samples, and insulation disruption can only be a result of SVs. Still, previous "pan-cancer" analyses showed that <14% of TAD boundary deletions actually result in a > 2-fold increase in gene expression of adjacent loci[23]. Thus, we exploited our high-resolution Hi-C data and a large number of samples assayed to uncover a key result of GBM rearrangements: the formation of hundreds to thousands (median = 120) of neoloops in patient genomes. Genes associated with neoloops were not only significantly higher expressed compared to counterparts without neoloops (thus, explaining intra-patient heterogeneity), but also enriched for genes characteristic of the GBM transcriptional program. Moreover, some recurrence in neoloop-associated genes could be observed (e.g., *HLA-J* overexpression associated with neoloop formation in 8 out of 28 GSCs). Notably, a substantial fraction of these neoloops ( > 35%) ectopically linked gene promoters with active enhancers leading to their activation in a tumor-specific manner. In fact, the formation of such potentiating neoloops can explain the overexpression of known GBM drivers, like *EGFR* and *MTOR*, in GSCs in cases where gene amplification or fusion does not. Having established how GBM inter-patient heterogeneity extends to and is supported by 3D genome refolding, we can envisage patient-specific vulnerabilities to be uncovered on the basis of 3D genomics analyses that might be able to inform personalized treatment decisions. We exemplify this by mapping and modeling 3D chromatin neo-structures in the *MYC* locus leading to selective sensitivity of a single GSC to a small molecule inhibitor[59].

Finally, as GBM relapse is essentially inevitable and the major hurdle in prolonging patient survival, studies have compared the genomic landscape of primary versus relapse *IDH*^wt GBMs. However,

they have reached contrasting conclusions. For example, Körber and colleagues[63] studied 21 primary-relapse tumor pairs using deep WGS to conclude that most of tumor evolution (including mutational selection) occurs prior even to primary diagnosis and, thus, relapse tumors share an overall similar landscape. This contrasts earlier work[64] and clinical experience, whereby GBM recurs tumultuously within a few months, and relapse tumors show little genetic resemblance to primary ones. One explanation for this disparity could be the local versus distal regrowth of tumors that seems to correlate with higher versus lower genetic resemblance[63,64]. Here, we studied three primary-relapse GSC pairs that recurred locally but in three different brain regions (i.e., occipital, frontal, parietal), following the exact same chemo- and radiotherapy regime. Our data on SV distribution and neoloop formation in each pair rather argue for reduced similarity. For instance, despite a consistent increase of SVs in relapse GSCs, there was an equally consistent loss of primary-specific SVs in relapse genomes. This can be explained by the two entities belonging to different (or very early diverging) tumor evolution trajectories. In addition, as our samples represent the stem cell-like compartment of GBM tumors, this could also mean that different GSC populations emerge after primary tumor resection and therapy that give rise to relapse tumors with divergent characteristics and resistance. We hypothesize that the formation of such dynamic 3D structures as neoloops is a means for expanding regulatory options in tumor cells and that neoloops are equally subject to tumor evolution as classical genomic alterations (e.g., amplifications and deletions) as they can support significant transcriptional effects. As a result, high SV dissimilarity may be less telling than high neoloop dissimilarity, as the latter can directly affect gene expression. Hence, relapse GSCs do diverge significantly from primary ones as regards their loop-level regulatory landscape despite local reemergence.

In summary, our Hi-C data constitute a valuable resource for GBM and exemplify how 3D genomics can be used to dissect patient-specific chromosomal effects. These can, in turn, help improve our understanding of GBM progression and rationally identify hitherto unknown prognostic markers and therapeutic vulnerabilities in the face of pervasive heterogeneity.

## Methods

### GSC generation and cell culture
Research reported here complies with all relevant national and international regulations, including the Declaration of Helsinki. The collection and processing of all samples was approved by the Ethics Board of the University Hospital, Catholic University of Rome (Prot. ID CE 2253), with informed consent obtained from all GBM patients. GBM tumors from 24 patients who underwent surgery at diagnosis ($n = 11$) or relapse ($n = 17$, as 3 initially-resected patients were also part of the relapse group) and standard radiochemotherapy at the Institute of Neurosurgery, Catholic University of Rome, were used to produce 28 GBM stem-like cell (GSC) lines. The key inclusion criteria were a GBM diagnosis (at first diagnosis or relapse; WHO Grade IV glioma), good patient functional status (Karnofsky score >70), often followed a standard Stupp protocol (for detailed profiles see Supplementary Table 1). Patient recruitment and study strategy was indiscriminatory towards biological sex, which was self-reported. We do not present data disaggregated for sex and gender for most analyses (SV detection is presented on a patient-by-patient basis in Supplementary Fig. 2) as the study design and patient numbers would not be sufficient for this (only 7 female GBM patients were recruited out of a total of 24).

For GSC generation, resected specimens were mechanically dissociated, and each resulting cell suspension was cultured in serum-free DMEM/F12 medium (ThermoFisher Scientific) containing 2 mM L-glutamine, 0.6% glucose, 9.6 mg/mL putrescine, 6.3 ng/mL progesterone, 5.2 ng/mL sodium selenite, 0.025 mg/mL insulin, 0.1 mg/mL transferrin sodium salt (Sigma Aldrich), human recombinant epidermal growth factor (hEGF; #AF-100-15, Peprotech; 20 ng/mL), basic fibroblast growth factor (b-FGF; #100-18B, Peprotech; 10 ng/mL) and heparin (2 mg/mL; Sigma Aldrich) at 37 °C under 5% $CO_2$. Proliferating cell cultures typically require 3 to 4 weeks to be established. GSCs were validated by Short Tandem Repeat (STR) DNA fingerprinting using nine highly polymorphic STR loci plus amelogenin (Cell ID™ System, Promega Inc). All GSC profiles were queried in public databases to affirm authenticity[65], and their in vivo tumorigenic potential was assayed by intracranial cell injection into immunocompromised mice, resulting in tumors with the same antigen expression and histological tissue organization as the tumor of origin[66,67]. Please refer requests for GSCs to R.P. or L.R.-V.

### MYC inhibition and immunofluorescence experiments
For MYC inhibition experiments, GSCs #148 (MYC^high) and #62 (MYC^low) were grown as described above, but using cell culture dishes coated with growth factor-reduced Matrigel (Corning) and dissociated using Accutase (Thermo Fisher) for passaging. Once expanded, GSCs were seeded on 12 mm sterile coverslips placed in each well of a 24-well tissue culture plate. Cells were treated with either 50 μM of the small molecule inhibitor EN-4 (Selleckchem)[59] or with an equivalent volume of DMSO for 48 h. All experiments were performed in at least three independent biological replicates. Following drug treatment, media was aspirated, and the cells were fixed in 4% paraformaldehyde (PFA) for 1 h at room temperature. Cells were next permeabilized in 0.5% Triton X-100 in PBS for 10 min and blocked with 0.5% fish gelatin in PBS for 1 h at room temperature.

MYC- and Ki67-positive cells were evaluated via immunofluorescence. In brief, primary antibody stainings were at 4 °C overnight, followed by 3×5-min PBS washes. Then, coverslips were incubated with anti-rabbit fluorophore-conjugated secondary antibodies (Sigma Aldrich) for 1 h at room temperature. The two primary antibodies used were rabbit polyclonal anti-Ki67 (Cat. No. AB9260, Merck Millipore; 1:1000 dilution) and rabbit polyclonal anti-MYC (Cat No. 10828-1-AP, Proteintech; 1:1000 dilution), while nuclei were also counterstained using DAPI. For TUNEL stainings, the DeadEnd™ Fluoremetric TUNEL kit (#TB235, Promega) was used according to the manufacturer's instructions. Finally, coverslips were mounted, and three raw images from random fields of view per coverslip were acquired using a Leica SP8 scanning confocal microscope (20x or 63x objective). Maximum intensity projection images were used, and MYC mean fluorescent intensity or the percentage of TUNEL-, Ki67-, and MYC-positive cells in each sample was computed using ImageJ.

### In situ Hi-C and data processing
Hi-C was performed on 0.5-1 million cells from each GSC line using the Hi-C+ kit (Arima Genomics) according to the manufacturer's instructions. Following sequencing on a NovaSeq platform (Illumina), Hi-C reads were aligned to the reference genome GRCh38 using *bwa mem* (v0.7.17) with "-SP5M". Invalid data, including PCR duplicates and read pairs mapping to the same restriction fragment, were removed using *pairtools* (v0.3.0)[68]. *runHiC* (v0.8.4-r1; https://zenodo.org/badge/doi/ 10.5281/zenodo) and *cooler* (v0.8.6)[69] were used to construct contact matrices at various resolutions. Raw Hi-C matrices were corrected using a modified matrix balancing method to account for CNV effects and other systematic biases including mappability, GC content, and restriction enzyme sites, all processed via *Neoloopfinder* (v0.3.0.post4)[24]. Stratum-adjusted correlation coefficients (SCC) between any two Hi-C contact matrices samples were calculated using HiCrep (v0.2.3)[70] at 10 kbp resolution.

PC1 was calculated, and A-/B-compartments identified at a resolution of 50 kbp using the *cooltools* (v0.3.2)[68] *call-compartment* function. Insulation scores and TADs were identified at 25 kbp resolution using the *cooltools* (v0.3.2) insulation function. Chromatin loops were

identified at 5-, 10-, and 25 kbp resolution on the basis of interaction probabilities > 0.95 and then merged using *peakachu* (v1.2.0)[71]. Significant differential loops were determined using the *diffPeakachu* function via the Gaussian mixture model of the *peakachu* probability score (FDR < 0.05). For 5- and 10 kbp loops, we extended anchors by 5 kbp when searching for associated TSSs to define loop-associated genes; for 25 kbp-resolution loops, no extension was applied.

To compare chromatin organization between GSC subtypes, hierarchical structural features (PC1, insulation scores, and loops) were used for unsupervised clustering of GSC samples with significant subtype enrichment scores. For PC1 and the insulation score, pairwise correlations were calculated per each genomic bin in all samples. For loops, differential loops between GSC pairs were identified as described above, and then Jaccard similarity indexes based on shared loops were calculated (we consider two loops as "the same" only if the midpoint of each anchor in one loop is within <50 kbp from the anchor midpoint in the other) before hierarchical clustering was performed on all matrices using average linkage and correlation distance metrics.

### Identification of structural variants (SVs) in Hi-C data

Structural variants, including inversions, deletions, duplications, and interchromosomal translocations, were detected and annotated using *EagleC* (v0.1.3)[29] on Hi-C data, which predicts SV breakpoints at single-kbp resolution and combines predictions from 5-, 10-, and 50 kbp resolutions. For 10- and 50 kbp predictions, EagleC further searches for the most probable local breakpoint coordinates within 5-kbp Hi-C contact maps so that all reported SVs are at the same resolution. In more detail, we divided the human reference genome (GRCh38) into 1 kbp bins and calculated a suite of metrics per bin to summarize a variety of properties with potential relevance to the distribution of SVs. To test for association between SV types and genome properties, each property was compared between SV breakpoint positions (randomly choosing one side of each breakpoint junction to reduce dependence between observations) and a set of 1000 randomly-shuffled SVs, keeping the SV breakpoint ends at the same distance and chromosome as those of bona fide ones. For each genome property and each SV type, real observations were pooled together with 1000 sets of random ones and rank-transformed and normalized on a 0-1 scale. Under the null hypothesis of no event-versus-property association, the ranks of the real observations would follow a uniform distribution. We tested this for each SV type using a Kolmogorov–Smirnov test with a Benjamini–Yekutieli FDR correction across the entire suite of tests and set the threshold for significance reporting at 0.01. To define duplicated and deleted genes induced by SVs, we used both orientation information of SV breakpoints and copy number profiles. Duplications were defined as intrachromosomal SVs with −+, ++, or −− orientations, and the genomic interval between breakpoints had a copy number ratio >1.35, while deletions were also defined as intrachromosomal SVs but with the +− orientation, and the genomic interval had a copy number ratio <0.65, considering allelic heterogeneity. Copy number profiles inferred from Hi-C were used in this calculation[24]. Local Hi-C maps surrounding SV breakpoints were reconstructed, and Hi-C signals across the breakpoints normalized due to the heterozygosity of the SVs and the potential heterogeneity of our patient-derived GSC samples. Then, neoTADs (predicted at 25 kbp resolution) and neo-loops (predicted at 5-, 10-, and 25 kbp resolutions with an FDR < 0.05 and then merged) on each local reconstructed map were detected. All steps were processed using *Neoloopfinder* (v0.3.0.post4). Finally, we used RNA-seq to identify fusion genes in all GSC samples using Arriba (v2.3.0)[72]. In parallel, we also used Hi-C processed via the EagleC (v0.1.3) *annotate-gene-fusion* function, as it can additionally detect intronic gene fusions[29]. In the end, fusion genes detected via both approaches were merged to provide a final list. All SVs, CNVs, neo-loops, and fusion genes are listed in Supplementary Data 1.

### RNA sequencing (RNA-seq) and data processing

GSCs grown to near-confluence in a T25 flask were directly lysed using Trizol (Invitrogen), total RNA was isolated using the DirectZol kit (Zymo), and used for standard poly(A)+ selection and library preparation with the TruSeq kit (Illumina). Following sequencing to at least 20 million reads on a NovaSeq platform (Illumina), reads were processed following the ENCODE pipeline (https://github.com/ENCODE-DCC/rna-seq-pipeline). Reads pairs were aligned to the human reference genome (GRCh38) and transcriptome (Gencode.v29) using STAR (v2.6.0c)[73]. Gene and isoform expression quantification was performed using RSEM (v1.3.3)[74]. Read coverage tracks (BigWig) were generated and normalized by scale factor using the *bamCoverage* function of deepTools2 (v3.5.1)[75]. Differentially-expressed genes were determined using RSEM (v1.3.3; *rsem-run-ebseq* function) with an FDR cutoff of <0.05. For the purpose of comparing expression levels across samples, we used "transcripts per million" (TPM) as the metric of choice and included all genes expressed at $\log_2(TPM + 1) > 0$.

For subtype classification, gene signatures from three different publications were used[51–53], and single-sample gene set enrichment analysis (ssGSEA) was conducted via R (ssGSEA). For each GSC, ssGSEA evaluated normalized enrichment scores for each signature set with TPM as input. Two-sided *P*-values of each sample were calculated by the corresponding normalized enrichment score via the *Z2p* package and used to determine the most significant subtypes for a given GSC expression profile.

### Cleavage Under Target and tagmentation (CUT&Tag) and data processing

GSCs were lifted from plates using accutase (Sigma-Aldrich). Typically, 0.5 million cells were processed using the CUT&Tag-IT kit (Active Motif) and 1 μg of anti-CTCF (Active Motif #61311) or anti-H3K27ac antibody (Active Motif #9133) as per manufacturer's instructions and the resulting libraries were sequenced on a NextSeq500 platform (Illumina) to obtain at least $10^7$ reads. Read pairs were aligned to the human reference genome GRCh38 using Bowtie2 (v2.3.4.1), PCR duplicates were removed using the *MarkDuplicates* function in Picard tools (v2.20.7), and read coverage tracks (BigWig) were generated and normalized with the RPCG parameter using the *bamCoverage* function of deepTools2 (v3.5.1)[75]. Peaks were called using SEACR (v1.3)[76] with an FDR cutoff of <0.01.

### Whole-genome sequencing (WGS) data processing

For WGS, read pairs were first mapped to GRCh38 by BWA mem (v0.7.17), and duplicate reads were removed by Picard (v2.20.7) as above. WGS-based CNV profiles and segments were calculated via the CNVkit (v0.9.9)[77] batch function using the "*--segment-method hmm-tumor -m wgs --drop-low-coverage --target-avg-size 25000*" parameters.

### TCGA data analysis

Kaplan-Meyer curves were generated via GEPIA2 (Gene Expression Profiling Interactive Analysis 2, v7.0)[78] based on 162 GBM samples from TCGA. Median gene expression values were used as a high-low group cutoff. Expression comparison between samples of glioblastoma and normal tissues were performed using GEPIA2 (v7.0) based on publicly-available TCGA and GTEx data.

### Simulations of SV impact on 3D genome folding

In order to predict neoloops forming as a result of translocations, we used a polymer physics-based approach previously used to predict ectopic interactions in *cis* arising in congenital disease-causing structural variations, PRISMR[60]. PRISMR models chromatin as a polymeric structure bearing sites of potential binding by proteins represented as floating particles in solution[79,80]. The thermodynamic properties of this model can be used to infer the binding site distribution along the polymer that best reproduces the Hi-C matrix of the genomic region

lacking the structural variation, while ectopic interactions are predicted by reshuffling the polymer in accordance with the variation and recalculating the new Hi-C contacts. Here, we modified the PRISMR approach to simulate neoloop formation by a translocation involving the *MYC* locus on chr8 and a large intergenic segment of chr12 (chr8: 127.71-127.78 Mbp; chr12: 57.7-58.155 Mbp); this occurs only in sample G148 of our cohort. Similar data from G394, where the translocation does not occur and *MYC* is inactive, provided a negative control.

To predict the best binding sites distribution for the hybrid region in G148, we first inferred them on an extended region on chr12 (i.e., chr12: 57.66-58.33 Mbp) using Hi-C data from G275R that carries no SVs across this segment. We then correlated the binding site distribution deduced from G275R with RNA-seq and H3K27ac CUT&Tag data from the same region in G148 to ensure good prediction of Hi-C contacts via a probabilistic approach using these correlations to extend binding site distribution prediction around the *MYC* locus on chr8 (i.e., chr8: 127.71-127.78 Mbp). Our approach repurposed PRISMR that finds the best minimum of the difference between the real Hi-C matrix and the reconstructed Hi-C matrix via a simulated annealing (SA) optimization procedure spanning the space of binding site distributions for a given number of classes. A contact matrix is then reconstructed via a mean-field approximation using contact probability profiles characterized in standard coil-globule polymer physics[60,79,80]. Estimation of the best number of binding site classes and the best λ (i.e., the regularization term used in PRISMR SA to penalize total binding site abundance and reduce overfitting) was as described[60]. In brief, SA was executed for a range of λ values, and the best λ was selected when the cost function raised ~10% above the starting plateau (Supplementary Fig. 7c). Similarly, SA was executed for an increasing number of binding sites classes, *M*, until the cost function did not show significant reduction (*M* = 11 was selected; Supplementary Fig. 7d). For this, experimental (input) Hi-C data was first smoothed via a Gaussian filter (0.5 bin), and similarity between the simulated and original contact matrices was estimated using distance-corrected Pearson's correlations (which were 0.65 and 0.45 for G148 and G394, respectively). Also, to account for chromatin persistence length effects in our 5 kbp resolution deduced Hi-C matrices, SA was applied independently for different monomer lengths by interpolating and scaling contact probability profiles accordingly, albeit with a significant increase in computing burden. To speed up optimization convergence, we modified SA to generate, at every iteration, multiple tentative modifications of the binding site configuration (rather than one in ref. 60) that were simultaneously evaluated. This allowed us to estimate an optimal monomer length ~20% longer than the 5 kb resolution.

To extend the prediction of binding site distribution to the *MYC* locus on chr8, we applied a probabilistic approach, using RNA-seq and H3K27ac data as a bridge between chromosomes. If $P_C$ is the probability of finding the binding sites class C in the region of interest, $P_T$ is the probability of finding the epigenetic track T, and *corr(C,T)* is the correlation between C and T, then the conditional probability *P(C | T)* to observe C given T can be obtained by inverting the following equations:

$$\mathrm{cov}(C,T) = P(C \cap T) - P_C P_T = P(C|T)P_T - P_C P_T \quad (1)$$

$$corr(C,T) = \frac{cov(C,T)}{(P_C(1-P_C)P_T(1-P_T))^{1/2}} \quad (2)$$

Here we define *corr(C,T)* as the correlation between the PRISMR-inferred best binding sites distributions (C) and the RNA-seq and H3K27ac tracks on the extended region chr12:57.66-58.33Mbp (T), and $P_C$ ($P_T$) as the frequency of observing C (T) in the same region. Once we have *P(C | T)*, we can estimate the probability of finding a binding site of class C in position x in region chr8:127.71-127.78Mbp from the

frequency of T as follows:

$$P_C(x) = \sum_{i \in RNAseq, H3K27ac} P(C|T_i) P_{T_i}(x) \quad (3)$$

In this formula, we neglected the intersection terms between $P_{RNAseq}$ and $P_{H3K27ac}$ as their correlation is quite low (< 0.2). When applying (3) we considered only (C,T) couples with a correlation of >0.2. Equations (1,2) follow from the very definition of correlation and covariance where $cov(C,T) = cov(\mathbf{1}_C \mathbf{1}_T)$ and $\mathbf{1}_X$ is the indicator function, so the expected values in the covariance equal the probabilities:

$$E(X) = p_X; \ \mathrm{var}(X) = p_X(1 - p_X) \quad (4)$$

## Prediction of contact maps for circular chromatin carrying an SV

To faithfully predict the contact map of the polymer carrying the SV, we modified the mean-field contact map reconstruction by assuming the circular topology of the chromatin fiber. We assumed that the linear distance between any pair of beads in the SV, used to assign the probability of contact, is given by the minimal distance within the circular structure of the fiber. Then, we validated this assumption with full 3D molecular dynamics simulations of the circular polymer.

For the *MYC* translocation, identification of the complete region is not straightforward, probably because of significant population heterogeneity. As a conservative assumption, we added an 80 kbp inert buffer region that closes the two flanks of *MYC* translocation and does not affect *MYC* locus dynamics if not for the circular topology induced. To validate the mean-field predicted circular topolog- corrected neoloops and get access to further quantities of interest, e.g., *MYC* expression, we informed real 3D polymer physics simulations with the circular topology of the translocation and with the predicted binding sites, as detailed in the next section.

## Simulation of 3D structure and dynamics of the reconstructed SV

To predict the 3D structure and dynamics of the genomic region bearing the translocation, we employed the SBS model via Molecular Dynamics simulations in a classical Langevin and velocity-Verlet framework with standard parameters[60,79,80]. The energy of interaction between binding sites and binders was set to 4 $K_B$T, while the binders' concentration was set to 200 nmol/liter. Randomly generated polymers and binder configurations were allowed to evolve and find the steady state before measuring the probability of contact. From the SBS predicted structures, we estimated the degree of *MYC* triplet colocalization with regions A and B with respect to what was expected by random independent pair-wise probability via the correlation coefficient:

$$corr(A,B) = \frac{P_{MYC,A,B} - P_{MYC,A} P_{MYC,B}}{\left(P_{MYC,A}(1 - P_{MYC,A}) P_{MYC,B}(1 - P_{MYC,B})\right)^{\frac{1}{2}}} \quad (5)$$

From the SBS polymer distance matrix we also estimated the level of *in-silico* single-allele *MYC* expression with respect to the average level F following the formula in ref.61:

$$\log\left(\sum_i d_{MYC,i}^{-1} / F\right) \quad (6)$$

where $d_{MYC,I}$ is the distance between *MYC* and *i* corresponding to a H3K27ac or RNA-seq peak.

## Statistics and reproducibility

All P-values were calculated using R, and their results were considered significant if *P* < 0.05 unless stated otherwise.

**Reporting summary**

Further information on research design is available in the Nature Portfolio Reporting Summary linked to this article.

## Data availability

Hi-C, RNA-seq, WGS, and CUT&Tag data not containing identifiable patient information have been deposited in the NCBI Gene Expression Omnibus (GEO) database under accession code GSE229966. Raw Hi-C, RNA-seq, and CUT&Tag data are protected by national policy and may be released for research use only upon request to A.P. (Email: argyris.papantonis@med.uni-goettingen.de) and approval by the UMG Ethics Board; the time frame for this will not exceed one month. GBM-associated genes used in this study are available in DisGenet v7.0 [https://www.disgenet.org/][48], data on CNVs of TCGA samples in the cBioPortal [https://www.cbioportal.org/], and cancer-related genes via the Bushman lab [http://www.bushmanlab.org/assets/doc/allOnco_May2018.tsv]. Astrocyte RNA-seq data[81] are available in the Sequence Read Archive under accession code SRP103788. Source data are provided in this paper.

## Code availability

All code used to analyze Hi-C, RNA-seq, WGS, and CUT&Tag data[82] is available at https://github.com/xieting0603/GBM; the custom code used to perform simulations[83] is available at https://github.com/marianoimperatore/MeanFieldChromatin.git.

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

## Acknowledgements

We thank Marieke Oudelaar (MPI-NAT) and Shiv K. Singh (UMG) for their critical reading of the manuscript and Kerstin Becker and Elizabeth Kirst from the Cologne Center for Genomics (CCG) for sequencing Hi-C libraries. This work was supported by the Italian Association for Cancer Research (AIRC; project IG2019 id. 23154 awarded to R.P.) and the

Ministero della Salute (project RF-2019-12368786 awarded to R.P.), the German Research Foundation (DFG) via the Clinical Research Unit 5002 (project 426671079 awarded to A.P.), the Priority Program SPP2202 (projects 422841138 and 422389065 awarded to R.F.S. and A.P., respectively), the Collaborative Research Center SFB1565 (project 469281184 awarded to A.P.), TRR81/3 (project 109546710 awarded to A.P.) and the individual grant PA 2456/15-1 (project 455784893 awarded to A.P.). R.F.S. is a Professor at the Cancer Research Center Cologne Essen (CCCE) funded by the Ministry of Culture and Science of the State of North Rhine-Westphalia and also received funds from the German Ministry for Education and Research (BMBF) as BIFOLD – Berlin Institute for the Foundations of Learning and Data (projects 01IS18025A and 01IS18037A). T.X. is supported by an Alexander von Humboldt Foundation postdoctoral fellowship; A.D-M., I.P. and N.Ü. are also members of the IMPRS Genome Science PhD program at the University of Göttingen.

## Author contributions

T.X. performed all computational analyses. A.D.-M. and I.P. generated Hi-C data. N.Ü. generated CUT&Tag and RNA-seq data. M.Ba. performed simulations. M.Bu., Q.G.D., L.R.-V., G.F. and L.L. generated all GSC lines. X.W. contributed computational code. O.S.V. and J.G. performed MYC inhibition experiments. R.F.S. and C.R. performed WGS analyses. T.X., R.P. and A.P. conceived the study. T.X. and A.P. compiled the manuscript with input from all co-authors.

## Competing interests

The authors have no competing interests to declare.
