## [Peer Review File · Nature Communications]

Pervasive structural heterogeneity rewires glioblastoma chromosomes to sustain patient-specific transcriptional programsEditorial Note: This manuscript has been previously reviewed at another journal that is not operating a transparent peer review scheme. This document only contains reviewer comments and rebuttal letters for versions considered at *Nature Communications* .

REVIEWER COMMENTS

Reviewer #1 (Remarks to the Author):

I thank the authors for addressing my and other reviewers' concerns. I have none remaining.

Reviewer #2 (Remarks to the Author):

This resubmission by Xie et al represents a strong improvement on what was already a very good manuscript. I really appreciate the effort the authors invested in this revised manuscript. Most of my comments were adequately addressed. I only have a couple of minor comments left.

- Regarding the comparison of EagleC and hicbreakfinder: Just because EagleC infers more SVs than hicbreakfinder does not mean that EagleC is better at reaching the ground truth. I think if the authors do not want to generate more WGS data to validate their SV predictions (my previous point in my initial appraisal of this manuscript), they should at least use two SV callers. Please remember that all good genomic papers on SVs and SNVs in cancer use multiple callers for the analyses of their WGS data. The use of complementary approaches is the norm in genomics, because at the end of the day, no computational package is perfect.

- The additions to Figs 2 and 3 are very strong. A small note here that the label on the y axis of Fig 2i has a missing r in what should be "other."

Reviewer #4 (Remarks to the Author):

In the previous version of the manuscript, reviewer 3 had four major concerns:

1. That genetic differences between individual patients could be more responsible for architectural differences than the cancers themselves.
2. That CNVs, and their effects independent of genome architecture, were overlooked in the analyses.
3. That the Hi-C analyses was not sufficiently benchmarked to show robustness.
4. That some claims were overstated.

Reading the original and revised manuscripts, I do not find either concerns 1 or 3 to be justified in the first place. As the authors point out, there is a lot of precedent for Hi-C maps to be overwhelmingly similar between healthy individuals (point 1), and that they have used the best existing analytical methods, benchmarked both in this manuscript and extensively in the literature (point 3).

The authors have now incorporated CNV analysis to tackle point 2, and have found that their major conclusions have not changed very much (with specific exceptions, for example at Myc, which have now been highlighted). They have also clarified their reasoning for which numbers are presented in each particularly analysis, and toned down the more contentious comments, dealing with point 4. Thus I recommend this manuscript for publication in Nature Communications.

Response to the Reviewers' Comments

Reviewer #1:

I thank the authors for addressing my and other reviewers' concerns. I have none remaining.

We thank the reviewer for the endorsement.

Reviewer #2:

This resubmission by Xie et al represents a strong improvement on what was already a very good manuscript. I really appreciate the effort the authors invested in this revised manuscript. Most of my comments were adequately addressed. I only have a couple of minor comments left.

We are happy to see that the reviewer acknowledges our efforts for this revision and the fact that they constitute “a strong improvement” to our “already very good” manuscript. The two minor points raised below are also now addressed.

- Regarding the comparison of EagleC and hicbreakfinder: Just because EagleC infers more SVs than hicbreakfinder does not mean that EagleC is better at reaching the ground truth. I think if the authors do not want to generate more WGS data to validate their SV predictions (my previous point in my initial appraisal of this manuscript), they should at least use two SV callers. Please remember that all good genomic papers on SVs and SNVs in cancer use multiple callers for the analyses of their WGS data. The use of complementary approaches is the norm in genomics, because at the end of the day, no computational package is perfect.

We do agree that no computational package is perfect. In response to this request from the reviewer, we have now added analysis of our Hi-C data at all levels using *hicbreakfinder* too [see new panels in **Fig. 5a,b** (comparison of primary-relapse GSC pairs) and in **Extended Data Figs 3a-c** (general SV and fusion gene detection), **5b-d** (neoTAD analysis), **6a-c** (neoloop analysis), **9a-c** (comparison of primary-relapse GSC pairs) and **10a,b** (comparison of center- versus periphery-derived GSCs)]. These additions are in agreement with all previous conclusions of the manuscript, but they do provide a more complete view of patient-specific GBM landscapes to the reader, which is a welcome addition to our work. New text describing the additions is marked in red in the main text.

- The additions to Figs 2 and 3 are very strong. A small note here that the label on the y axis of Fig 2i has a missing r in what should be “other.”

We thank the reviewer for noticing this mistake, which we have now corrected.

Reviewer #4:

In the previous version of the manuscript, reviewer 3 had four major concerns:

- 1. That genetic differences between individual patients could be more responsible for architectural differences than the cancers themselves.*
- 2. That CNVs, and their effects independent of genome architecture, were overlooked in the analyses.*
- 3. That the Hi-C analyses was not sufficiently benchmarked to show robustness.*
- 4. That some claims were overstated.*

Reading the original and revised manuscripts, I do not find either concerns 1 or 3 to be justified in the first place. As the authors point out, there is a lot of precedent for Hi-C maps to be

overwhelmingly similar between healthy individuals (point 1), and that they have used the best existing analytical methods, benchmarked both in this manuscript and extensively in the literature (point 3). The authors have now incorporated CNV analysis to tackle point 2, and have found that their major conclusions have not changed very much (with specific exceptions, for example at Myc, which have now been highlighted). They have also clarified their reasoning for which numbers are presented in each particularly analysis, and toned down the more contentious comments, dealing with point 4. Thus I recommend this manuscript for publication in Nature Communications.

We thank the reviewer for putting our arguments in context and for their endorsement.

REVIEWERS' COMMENTS

Reviewer #2 (Remarks to the Author):

The authors addressed all my concerns.

Response to the Reviewers' Comments

Reviewer #2:

The authors addressed all my concerns.

We thank the reviewer for the endorsement.